# Development of a Xanthan Gum Based Superabsorbent and Water Retaining Composites for Agricultural and Forestry Applications

**DOI:** 10.3390/molecules28041952

**Published:** 2023-02-18

**Authors:** Alessandro Sorze, Francesco Valentini, Andrea Dorigato, Alessandro Pegoretti

**Affiliations:** Department of Industrial Engineering and INSTM Research Unit, University of Trento, via Sommarive 9, 38123 Trento, Italy

**Keywords:** xanthan gum, cellulose fibers, hydrogels, forestry, soil conditioner, topsoil cover

## Abstract

In this work, bio-based hydrogel composites of xanthan gum and cellulose fibers were developed to be used both as soil conditioners and topsoil covers, to promote plant growth and forest protection. The rheological, morphological, and water absorption properties of produced hydrogels were comprehensively investigated, together with the analysis of the effect of hydrogel addition to the soil. Specifically, the moisture absorption capability of these hydrogels was above 1000%, even after multiple dewatering/rehydration cycles. Moreover, the soil treated with 1.8 wt% of these materials increased the water absorption capacity by approximately 60% and reduced the water evaporation rate, due to the formation of a physical network between the soil, xanthan gum and cellulose fibers. Practical experiments on the growth of herbaceous and tomato plants were also performed, showing that the addition of less than 2 wt% of hydrogels into the soil resulted in higher growth rate values than untreated soil. Furthermore, it has been demonstrated that the use of the produced topsoil covers helped promote plant growth. The exceptional water-regulating properties of the investigated materials could allow for the development of a simple, inexpensive and scalable technology to be extensively applied in forestry and/or agricultural applications, to improve plant resilience and face the challenges related to climate change.

## 1. Introduction

In recent years, climate change has caused serious damage to forest ecosystems. Violent storms, wildfires, soil erosion, and the proliferation of insects and fungi have become very common phenomena, leading to the clearing of large areas of forests [1,2,3,4]. In addition, restocking and restoring forests are even more daunting tasks given the lack of rainfall in many parts of Europe [5,6]. Moreover, the southern regions of Europe have recently been exposed to a high risk of desertification, due to the overexploitation of land, soil, and water, urban sprawl, tourism, and unplanned industrialization [7,8]. To address these issues, there is a need for the development of technological solutions to support planting and silviculture through appropriate forestry operations, with the possibility of restoring even degraded lands. In the last decades, different methods have been employed to improve water use efficiency and thus to increase the survival rate of growing plants [9,10,11,12,13].

One of the most promising and effective tools is the use of soil conditioners (SCs), i.e., products that are mixed with the soil in the planting hole and are aimed to improve the chemical, physical, and water-regulating characteristics of the soil [14,15]. Superabsorbent hydrogels, capable of absorbing more than 300 times their dry weight, have been thus extensively investigated for water retention and soil conditioning applications [16,17,18,19,20,21,22,23,24,25]. Several studies showed that they can help lower water consumption during irrigation, decrease plant mortality rate, and thus increase the growth rate of trees [26,27,28]. Another interesting solution regarding the use of mulching films or topsoil covers (TSCs), is the use of layers that cover the seedling to improve the regulation of moisture evaporation from the soil and at the same time protect the plant from competing weeds [29,30]. However, soil conditioners and topsoil covers are commonly based on synthetic polymers (mainly polyacrylamide) and plastic films, respectively. These materials present heavy drawbacks related to their limited biodegradability and the release on the soil of their by-products, which are considered dangerous for the environment and for the human beings [16,31,32,33]. In order to overcome these problems, different biopolymers have been recently proposed to replace these materials [34,35,36,37,38,39,40].

Biopolymers are polymeric materials derived from renewable sources and are biodegradable [41,42,43,44,45]. They can be polysaccharides, such as cellulose, or proteins, such as gelatin, casein, and silk. They can be chemically synthesized from bio-derived monomers (e.g., polylactic acid) [46], or they can be produced from microbial activity (e.g., xanthan gum, poly-hydroxy alkanoates), but they can also be derived from fossil fuels, as long as they are biodegradable (e.g., polycaprolactone). Biopolymers are eco-friendly materials that have been widely employed in food and medical applications [47,48]. For soil conditioning purposes, xanthan gum is one of the most promising materials, due to its biodegradability, soil-strengthening efficiency, film-forming ability, and high water-absorbing capability [49,50,51,52,53]. It is a polysaccharide produced by Xanthomonas campestris through aerobic fermentation [54], composed by two glucoses, two mannoses, and one glucuronic acid unit that mostly form helical structures [55,56]. Xanthan gum is soluble in both cold and hot water and therefore can easily form hydrogels. The use of fillers, based on wood or cellulose fibers, may offer the possibility of optimizing the performances of the materials, in terms of water absorption, water retention, and structural stability [57,58,59,60].

Despite the possible synergies derived from the combination of xanthan gum with cellulose fibers to produce materials for soil conditioning and water retention, no study can be found in the open literature on this topic. Therefore, the aim of this work is to develop, for the first time, engineered bio-based composites constituted of xanthan gum and cellulose fibers to be used both as soil conditioners (SCs) and topsoil covers (TSC) for agricultural and forestry applications.

The first product is based on a hydrogel of xanthan gum and cellulose fibers, while in the second case, the same materials have been crosslinked with citric acid to produce bio-based biodegradable films. The choice of the materials and the methods adopted for the production of these composites have been optimized, so that they can be easily produced in large quantities for forestry and/or agricultural applications. The idea is to develop materials that are compatible with every type of soil (both sandy and clay), and that have better performances and comparable costs with products already present on the market.

## 2. Results and Discussion

### 2.1. Rheological Properties

In Figure 1a–c, the results for the rheological analysis performed on the hydrogel composites are displayed. In particular, in Figure 1c, the values of yield stress are displayed by fitting the viscosity data according to the Casson model (see Equation (1)).

Rheological measurements show that by increasing the amount of the xanthan gum and cellulose fibers in the composites, the viscosity and the shear stress are increased up to one order of magnitude (from 100 to 1000 Pa·s and from 1 to 10 Pa, respectively). Moreover, from Figure 1b, it can be observed that these composites show a yield stress, which indicates the minimum stress required for the material to flow. Figure 1c shows that the yield stress increases with the amount of xanthan gum and cellulose fibers in the solution. This means that the increase in the concentration of xanthan gum in the solution and the presence of the fibers leads to an increase in the ionic strength of the system and thus to the strengthening of the gel network, as also confirmed by Talukdar et al. [61] who investigated the drug release behavior of xanthan gum solutions.

### 2.2. FT-IR Spectroscopy

In Figure 2, the results for the FT-IR analysis performed on the topsoil covers and neat materials are shown.

From Figure 2, it is possible to observe that the characteristic spectra of xanthan gum (XG) are present in all of the TSC samples. In particular, the broad absorption peak around 3300 cm^−1^ corresponds to the O-H stretching vibration of the hydroxyl group, while the peaks at 1615 and 1408 cm^−1^ are associated with the asymmetrical and symmetrical C=O stretching vibration of carboxylate anion (-COO^−^), respectively. The characteristic peak at 1014 cm^−1^ for the C-O stretching of primary alcohols is also clearly observed. The peak at 1714 cm^−1^ is ascribed to the carbonyl groups of carboxylic acid (-COOH) [62]. Comparing the spectra of neat XG with those of TSCs, it can be seen that this peak is more evident by increasing the amount of crosslinking agent in the samples. Ester groups were formed from the crosslinking reaction between xanthan gum and citric acid and this leads to the overlapping of the absorption peaks of the carbonyl group from both ester and carboxylic acid at 1714 cm^−1^ [63]. In fact, for the TSC_B sample, which has the highest amount of citric acid, the intensity of this peak is stronger, meaning that ester groups are present in the structure, while for the TSC_D sample without citric acid, this peak is weaker than the others. The characteristic peaks of cellulose fibers (W) are similar to those of xanthan gum, due to their similar chemical structure and thus were difficult to distinguish in the TSC samples.

### 2.3. Light Microscopy

Figure 3a–i shows the light microscopy images for each hydrogel sample.

It is possible to notice that an increase in the concentration of xanthan gum and cellulose fibers in the solution leads to a densification of the gel structure, passing from a very aqueous solution (X1) to a more viscous and denser one (X4W5). Moreover, it can be observed that the structure appears to be quite homogeneous in all of the samples, without the presence of evident lumps and agglomerates in the composites. This means that the mixing process was performed efficiently with an effective dispersion of the polymer and the fibers in the water solution. It can be hypothesized that the network of cellulose fibers within the hydrogel structure may lead to a reinforcing and stabilizing effect of the material, when it is applied in the soil [64,65].

### 2.4. Water Absorption Properties

In Figure 4a,b, the results of the moisture absorption and water release measurements for the different hydrogel formulations are reported.

Figure 4a shows that the moisture absorption of the tested samples increases almost linearly with time and that the values of the moisture absorption capacity are higher for the samples with a lower amount of xanthan gum and cellulose fibers. The increase in the concentration of xanthan gum and cellulose fibers in the solution leads to a decrease in the moisture absorption capacity. This behavior can be explained by the chemical similarity between xanthan gum and cellulose fibers which may lead to a good interface adhesion between the two components, which, to some extent, can slow down the diffusion of water molecules [58]. Moreover, increasing the amount of biopolymers in the hydrogel may result in the presence of lumps that may adversely affect moisture absorption. The analysis was terminated before reaching the plateau due to the formation of molds in the samples, which formed because the test was performed under elevated humidity conditions. Figure 4b shows that the water loss mainly occurs in the first two hours of the test, with an opposite trend with respect to water uptake measurements: an increase in the amount of fibers leads to a slower water release. Furthermore, the formation of a fiber network in the hydrogel (see Figure 3) helps to limit the volumetric shrinkage during the drying, which is very important for samples without cellulose fibers. From these measurements, it can be concluded that the most promising formulations for practical application are those that couple a good water absorption capacity and a slow release of water, such as X1W2, X4W2 and X4W5 samples.

In Figure 5a,b, the results of water absorption and release tests for the xanthan gum-based top soil cover are shown.

From Figure 5a, it can be observed that the samples with a lower amount of citric acid (TSC_A and TSC_C) have a higher water absorption capability (up to 2000% after 700 h), compared to sample TSC_B, which has a higher citric acid content. It is also possible to observe that there are no evident differences between the results of samples TSC_A and TSC_C, meaning that the presence of cellulose fibers does not affect the water absorption tendency of the samples. However, from Figure 5b, it can be seen that the presence of fibers is important because they allow for a slower release of water, as observed in the TSC_A sample, compared to TSC_C. Furthermore, also in this case, the presence of the fibers in the hydrogel limits the volumetric shrinkage during drying. The water uptake, in this case, mainly occurs in the first hours due to the fact that, differently from hydrogels, the xanthan gum-based films were directly immersed in water. Moreover, the weight loss kinetic is similar.

### 2.5. Application on Soil

#### 2.5.1. Morphological Analysis of the Soil

In Figure 6a–f, the SEM micrographs of the soil and soil mixed with xanthan-based hydrogels are shown.

From the SEM micrographs of the neat soil shown in Figure 6a, it is possible to observe that soil particles are mainly distributed within two dimension ranges: the first with particles around 50 µm (47.8 ± 13.8 µm) and the second with particles around 10 µm (12.2 ± 4.3 µm). From Figure 6d and 6f, it is possible to notice that the cellulose fibers have a diameter of 7.9 ± 2.4 µm and an average length of 300 µm. Comparing Figure 6a with Figure 6b–f, it is possible to observe that mixing the soil with the xanthan gum-based hydrogels leads to an agglomeration of the soil particles, with the formation of an interconnected structure. The interconnection between the soil particles, xanthan gum, and cellulose fibers is even more evident by increasing the content of the biopolymer and filler mixed with the soil. This behavior can be probably explained by the capability of xanthan gum to make hydrogen bonds with the fine soil particles, as observed by Chang et al. [50]. Moreover, from these micrographs it is evident that the importance of cellulose fibers in forming an interconnected network may lead to beneficial effects in the case of real applications (e.g., soil stabilization, landside mitigation, and reduced water loss).

#### 2.5.2. Evaluation of the Maximum Water Holding Capacity (MWHC) of the Soil

In Figure 7, the results of the MWHC tests of the soil mixed with different hydrogels are reported. In this plot, a comparison with a commercial product (Idrogea) is also reported.

From Figure 7, it can be recognized that the addition of hydrogels to the soil increases its maximum water-holding capacity (MWHC) value. The MWHC of the reference untreated soil is 32.8%, meaning that the addition of 33 mL of water leads to the saturation of 100 g of soil. The soil treated with the dried and pulverized X4W2_low and X4W2_high compositions, allows the MWHC values to increase by 20.5% and 61.6%, respectively. In particular, the X4W2_high sample reaches a maximum water holding capacity as high as 53.1%, much higher with respect to the soil treated with the commercial product (Idrogea), that shows an MWHC value of 41.1%, comparable with the value obtained for the X4W2_low sample. Moreover, the MWHC value could not be increased by further increasing the amount of the commercial product in the soil, because it would have resulted in a relevant increase in the soil volume, with consequent geological instability in practical applications.

#### 2.5.3. Evaluation of the Water Retention Capacity of the Soil

In Figure 8, the results of the evaluation of the water retention capacity of the soil are reported.

Figure 8 shows that adding hydrogel to the soil leads to an increase in the water retention capacity, which means a slowdown in the water evaporation rate from the soil. The neat soil has a weight loss of 71.8% after 8 days, and became completely dry after 14 days. The soil treated with the two compositions of X4W2 possesses weight loss values of 71.4% (low) and 56.5% (high) after 8 days. In these cases, complete dryness is reached after 14 and 24 days, respectively. The addition of the commercial product Idrogea leads to a weight loss of 68.5% after 8 days and to complete drying after 14 days. Therefore, also from this test, the xanthan-based hydrogels show a better performance in retaining water and thus are a more suitable solution for soil conditioning applications, acting as a micro water reservoir for plants.

#### 2.5.4. Case Study Application

Evaluation of the plant growth rate with the soil conditioner

In Figure 9a–d, the results of the first plant cultivation experiment with soil conditioners are reported.

From Figure 9b, it can be noticed that, after 18 days, the grass is fully grown in all types of culture media, which means that the selected materials are not harmful to plants. In addition, Figure 9c shows that the presence of hydrogel composites does not affect the initial germination of the seeds and the overall growth of sprouts. Indeed, the germination rate, defined as the ratio of the number of germinated seeds to the number of planted seeds, is similar for each pot. This means that these composites are not thought to be used to promote seed germination rate but are more suitable to support the growth of already hatched seedlings. From Figure 9d, it can be observed that the average plant height is unaffected by the presence of the composites; in particular, the overall average plant height after 18 days is 9.7 ± 1.7 cm. Figure 10 shows an image of the grass after 7 days of the interruption of the water supply.

From Figure 10 it can be observed that, after 7 days of drought conditions, the grass is still lush for some specific formulations. In particular, the soil mixed with X1W2, X4W2, and X4W5 hydrogels shows the best results in terms of grass survival rate. As already reported in Figure 4a,b, these three hydrogel formulations show a good balance between a good moisture absorption capacity and slow water release, with consequent beneficial effects in case of a prolonged absence of water supply.

However, for X1W2 and X4W5 formulations, mold has formed on the top layer of the soil. Mold growth in these systems can be explained by the high concentrations of cellulose fibers with respect to the amount of xanthan gum, which can induce accelerated biodegradation of the composite with consequent proliferation of bacteria and fungi, especially at elevated humidity conditions. The X4W2 sample possesses the optimized ratio of xanthan gum and cellulose fibers, demonstrating the applicability of this hydrogel composition as a soil conditioner to support grass growth in drought conditions, without the formation of molds.

In Figure 11a–d, the results for the second plant cultivation experiment with soil conditioners are reported.

From Figure 11c, it is possible to observe that, in this case, the germination rate for the culture medium consisting of the soil treated with X4W2 is two times that of the soil treated with the commercial product. From Figure 11d, it can be seen that the average plant height is very similar for each sample, in particular, the overall height is 4.7 ± 1.7 cm and 9.1 ± 2.1 cm, after 7 and 18 days, respectively, which is very similar to those of the previous experiment. From these studies, it can be concluded that the presence of xanthan gum and cellulose fibers in the soil helps to better retain water, which is delivered to the plant more gradually than in the neat soil, giving plants the chance for better survival rates under water scarcity conditions. The presence of the hydrogel improves the water retention of the soil due to the hydrophilic characteristics of its constituents, providing a suitable environment for plant cultivation.

Evaluation of the Plant Growth Rate with Topsoil Cover

In Figure 12a,b, the representative pictures of the plant cultivation with TSC at two different times are shown.

Figure 12a,b clearly show the effect of the TSC on plant growth. It is possible to observe that the two plants with the TSC (1 and 2) have a greater vegetation mass than the plant taken as a reference (3). This indicates that, in addition to not being harmful to plants, the selected TSC formulations help to promote plant growth and improve the water regulating properties of the soil by slowing down the evaporation rate.

In Figure 13a,b, the visual aspect of the TSC after 6 days and 80 days is displayed.

From Figure 13a,b, it is possible to observe that after 80 days the TSC undergoes evident biodegradation with a volumetric shrinkage that is more significant for sample 1 (not crosslinked TSC), which means that heat treatment is useful for increasing the stability and the strength of the TSC. However, in both cases, the TSC works efficiently for a period of time that is comparable with the average lifespan of a tomato plant (2–3 months) making it suitable for this type of application. Moreover, it is important to point out that water was supplied to the plant every day, which resulted in the TSCs being subjected to many absorption/drying cycles that greatly stressed the material with consequent accelerated biodegradation. For other applications, such as in forestry, in which the service life of a TSC is expected to be longer, it is important to note that the water supply would be determined only by the weather conditions and may not be supplied on a daily basis. In these conditions, the TSC would be less stressed by hydration/dehydration cycles, and its service life would be therefore increased. However, it is clear that the composition of these TSCs should be optimized, improving their durability and weathering stability for a period of at least 24 months, in order to make them suitable for real agricultural and forestry applications. Finally, it can be also observed from Figure 13b, that the presence of the TSC on the soil helps to partially block the growth of competing weeds, showing that the presence of a durable and stable TSC on the soil can be effective as expected, in this sense as well.

## 3. Materials and Methods

### 3.1. Materials

Commercial xanthan gum, with a purity > 91% and molecular weight (M_w_)~1.0 × 10^6^ g/mol, was purchased from Galeno srl (Prato, Italy) in the form of fine powder and used as received. Natural cellulose fibers (Arbocel grade R) were kindly provided by J. Rettenmaier & Söhne Gmbh (Rosenberg, Germany) in the form of powder with an average fiber length of 200–300 µm and bulk density of 60–105 g/L [66]. Citric acid monohydrate, with a purity 99.5–101% and M_w_~210.14 g/mol, was supplied by Riedel-de Haën GmbH (Seelze, Germany) and used as a crosslinking agent.

The soil was sampled from a forest near the city of Trento (Italy). Appropriate analyses for the characterization were carried out by Fondazione Edmund Mach (San Michele all’Adige, TN, Italy), and its main properties are reported in Table 1. As a benchmark, tests on the soil were also performed using a commercial potassium polyacrylate-based soil conditioner, called Idrogea and purchased from Endofruit srl (Verona, Italy). According to the technical datasheet, the concentration of this product to be used in the soil is 0.13 wt%.

### 3.2. Sample Preparation

For the preparation of the soil conditioners, hydrogels were produced by dissolving the xanthan gum powder in water at different relative concentrations (1 wt%, 2 wt% and 4 wt%). This biopolymer is soluble in cold water, so the procedure was carried out at room temperature by mixing the solution with the use of a Dispermat^®^ F1 mixer (VMA-Getzmann Gmbh, Reichshof, Germany), operating at 5000 rpm for 15 min, in order to obtain a homogeneous mixture without lumps. During mixing, different amounts of cellulose fibers (0 wt%, 2 wt% and 5 wt% relative to water solution) were gradually added. In this way, nine different compositions were obtained, as reported in Table 2.

For the preparation of the topsoil covers, xanthan-based films were produced following the method described by Bueno et al. [67]. Citric acid was added to the 4 wt% xanthan gum solution, and after homogenizing it with the mixer for 3 min at 10,000 rpm, the solution was cast and dried at 45 °C overnight. The crosslinking reaction was performed by heating the dried films in an oven at 165 °C for 7 min. The reaction scheme of the crosslinking is reported in Figure 14. In this case, four different samples were produced by tuning the amount of citric acid (0 wt%, 1 wt% and 2 wt% relative to the biopolymer matrix) and the amount of cellulose fibers (0 wt% and 2 wt%), as reported in Table 3.

### 3.3. Characterization

#### 3.3.1. Rheological Properties

The rheological properties of the hydrogel samples were investigated using a Discovery hybrid rheometer DHR-2 (TA Instrument, DE, USA), adopting a plate-plate configuration and setting a gap distance of 1 mm. The tests were performed under steady conditions at a constant temperature of 30 °C, in order to measure the viscosity and the shear stress in a range of shear strain between 0.1 to 100 s^−1^. From these tests, it was possible to determine the yield stress of the materials by fitting the data with the Casson model [68], which can be described by Equation (1):(1)$$\sigma^{1/2}=\sigma_{y}^{1/2}+{\left(\eta \dot{\gamma }\right)}^{1/2}$$
where *σ* is the shear stress, *σ_y_* is the yield stress, *η* is the shear viscosity and $\dot{\gamma }$ is the shear rate. The value of the yield stress was detected as the intercept with the *y*-axis of the linear fit of the data in the strain rate range from 0.3 to 2 s^−1^.

#### 3.3.2. FT-IR Spectroscopy

Infrared spectroscopy (FT-IR, Fourier-transformed infrared spectroscopy) tests were conducted in attenuated total reflectance (ATR) mode using a PerkinElmer Spectrum One spectrometer (PerkinElmer, Waltham, MA, USA), in the wavelength number range of 650–4000 cm^−1^, obtaining each spectrum from the superposition of four scans. The analyses were performed on topsoil cover samples in order to investigate the effect of the crosslinking agent. Moreover, neat xanthan gum (XG) and cellulose fibers (W) were tested as reference.

#### 3.3.3. Light Microscopy

The different compositions of hydrogel composites in the wet state were observed using a Nikon SMZ25 light microscope, equipped with a Nikon DS-Fi2 digital camera. This analysis was performed to study the morphology of the prepared samples and to investigate the presence of agglomerates in the solutions, assessing thus the feasibility of the mixing process.

#### 3.3.4. Water Absorption Properties

Moisture absorption analyses were carried out on the hydrogel samples previously dried for 48 h in an oven at 50 °C. The analysis was carried out by periodically weighting the samples stored in a sealed box under constant humidity (RH = 80%) and temperature (25 °C) conditions. The mass of the samples was measured with a Gibertini E42 balance (resolution of 0.1 mg). The water uptake (WU%) was calculated according to Equation (2):(2)$$WU\%=\frac{m_{wet}-m_{dry}}{m_{dry}}\cdot 100$$
where *m_wet_* is the mass of the sample in the humid environment and *m_dry_* is the initial sample mass after drying. Following the complete hydration of the samples, they were placed in an oven at a constant temperature of 40 °C. In this way, by periodically measuring the weight of the samples, it was possible to calculate the weight loss (WL%), which indicates the water evaporation rate, according to Equation (3):(3)$$WL\%=\frac{m_{i}}{m_{wet}}\cdot 100$$
where *m_i_* is the mass of the sample during drying. In the case of the TSC samples, the water-holding capacity was evaluated by immersing dried specimens in demineralized water for a certain amount of time, following ASTM D570-98 standard and using Equation (2). For this analysis, only the crosslinked samples (TSC_A, TSC_B and TSC_C) were considered, because they kept their shape and consistency when immersed in water, while the not crosslinked sample (TSC_D) was completely dissolved in water after a few days. Furthermore in this case, the weight loss of the fully wet samples, which corresponds to the water evaporation rate was evaluated placing samples in an oven at a constant temperature of 40 °C, using Equation (3).

#### 3.3.5. Application on the Soil

To evaluate the applicability of the produced hydrogels as soil conditioners, the morphological properties, water-holding capacity, and water retention of the soil treated with hydrogels were investigated. The analyses were performed only on the soil with the X4W2 sample, that, from the tests carried out in the first part of this work, showed the most interesting moisture absorption and water retention capacities, good rheological properties, and mold resistance. Specifically, the tests were performed on the soil mixed with two different concentrations of dried X4W2 hydrogel: X4W2_high (1.8 wt% relative to soil) and X4W2_low (0.45 wt% relative to soil).

##### Morphological Analysis of the Soil

Micrographs of the soil mixed with hydrogel samples were obtained by using a field emission scanning electron microscope (FESEM) Zeiss Supra 40, operating at an accelerating voltage of 3.5 kV inside a chamber under a vacuum of 10^−6^ Torr. Soil samples were mixed with the prepared hydrogels in order to have a water-soil ratio of 30%. Prior to the observations, the samples were dried in a fan oven at 50 °C for 48 h and then coated with a thin electrically conductive Pt/Pd layer.

##### Evaluation of the Maximum Water Holding Capacity (MWHC) of the Soil

The determination of the water holding capacity was performed on the soil treated with hydrogels to assess the efficiency of these bio-composites as soil conditioners. These tests were carried out on samples of soil filtered through a 1.8 mm sieve and then mixed with X4W2. A soil sample mixed with the commercial product Idrogea was analyzed for comparison, and untreated soil was also tested as reference. A fixed amount of soil (100 g) was kept in a holed plastic beaker, whose base was covered with filter paper. The beakers were weighted (m_dry_) and then water was poured from the top of the beaker until the saturation limit of the soil was reached (i.e., until water seeped out from the bottom or a meniscus of water formed above the soil). At saturation, the samples were weighted again (m_wet_) and the MWHC was calculated using Equation (2). Three specimens were tested for each composition.

##### Evaluation of the Water Retention Capacity of the Soil

The effect of the presence of hydrogel in the soil on the water retention capacity was evaluated by adapting the procedure utilized by Ni et al. [69]: 100 g of dry soil were filtered through a 1.8 mm sieve and mixed with the dried hydrogels. One hundred mL of tap water were then slowly added to the samples and weighed. Neat soil was also tested as reference. The samples were maintained at 30 °C in an oven and periodically weighed. The weight loss which corresponds to the water evaporation rate was calculated using Equation (3).

##### Case Study Application

Evaluation of the plant growth rate with soil conditioner

The produced xanthan gum-based hydrogels were applied to the soil to investigate the effects of the biopolymers on the plant growth. Two different experiments were carried out. In the first experiment, the soil mentioned in Section 3.1 was mixed with the nine different compositions of soil conditioner reported in Table 2. The mixing conditions were the same described in Section 3.2 and the experimental procedure was adapted from a work of Chang et al. [28]. Each culture media was distributed in three pots (length 4 cm × width 4 cm × depth 5 cm) and three more pots for the neat soil were used as a reference, as shown in the schematization represented in Figure 15. In each of the 30 pots, the same number of grass seeds of *Nepeta Cataria* (HortuSì Srl, Italy) were planted. The seeds were covered with a thin layer (i.e., 5–10 mm) of soil and placed in a sealed chamber under constant thermo-hygrometric conditions (21 °C; 30%). Only 10 mL of water were supplied daily in each pot and no nutrients were applied. Seed germination of each soil/medium type was monitored periodically. Then, after 20 days, the daily water supply was cut off and the analysis was focused on which plants better survived in dry conditions.

In the second experiment, the same analysis was carried out by comparing the soil mixed with X4W2 hydrogel, that showed the best results from the previous analysis, with the soil mixed with the commercial product.

Evaluation of the Plant Growth Rate with Topsoil Cover

For the topsoil cover, a preliminary practical experiment was carried out by evaluating the growth of tomato plants (*Solanum lycopersicum var. cerasiforme* purchased by Consorzio Agrario Trento, Italy) to study the possible interactions of the topsoil cover in the soil and with plants. Monitoring continued for 80 days, comparing the increase in vegetation mass of plants and observing the biodegradability of TSC in the soil. Three tomato plants, 30 cm in height, were planted at a distance of 80 cm from each other in a common garden soil. Six hundred mL of water were provided daily to the plants and no fertilizers were used. Xanthan gum films, with dimensions of 50 × 50 cm^2^, were used as topsoil covers and applied at the base of two plants, while the third was planted without TSC and used as a reference. Specifically, the two topsoil covers utilized had the composition of TSC_A, but only one of them was subjected to heat treatment in a furnace to perform the crosslinking reaction. This made it possible to verify the effectiveness of this treatment and to evaluate the influence of the crosslinking step for these materials in a practical case study.

In Figure 16, the experimental setup utilized for the tomato plant cultivation is displayed. The three plants were numbered from 1 to 3. The first was planted with TSC_A not crosslinked in an oven, the second was planted with TSC_A crosslinked in an oven and the third one was the reference, planted without TSC.

## 4. Conclusions

This work demonstrated that novel biodegradable hydrogel composites produced from xanthan gum and cellulose fibers could be effectively used to develop soil conditioners and topsoil covers capable of promoting plant growth and forest protection. The xanthan gum biopolymer was very suitable for these applications due to its efficient water-absorbing properties. The addition of natural fillers, such as cellulose fibers, resulted in an increased water retention capability, thanks to the formation of a physical network between the soil and xanthan gum. For the application as a soil conditioner, several formulations of hydrogels were produced by adjusting the amount of xanthan and cellulose fibers. Among them, the composition with 4 wt% of xanthan gum and 2 wt% of cellulose fibers (X4W2) showed interesting moisture absorption and water retention capacity, good rheological properties and mold resistance. The addition of 1.5 wt% of X4W2 hydrogel to the soil improved the ability to absorb water by 60% with respect to the neat soil, and it also increased the water retention by slowing the rate of water evaporation from the soil. This formulation displayed better absorption and water retention performances with respect to a commercial potassium polyacrylate-based soil conditioner.

For the topsoil cover application, the crosslinking of xanthan gum with citric acid allowed the TSC to absorb water while maintaining its dimensional stability and durability in the soil. From practical experiments conducted on the growth of herbaceous and tomato plants, it was observed that the presence of xanthan gum-cellulose fiber composites in the soil helped to better retain water, which was released to the plant more gradually than in untreated soil, thus enabling a more vigorous growth and an increased survival rate under drought conditions. Further studies are underway to optimize the formulation of TSC, in order to increase its durability in the soil and allow an easier production process. In addition, practical applications on forests are already being planned to test the performances and feasibility of these materials for this application.

## Figures and Tables

**Figure 1 molecules-28-01952-f001:**
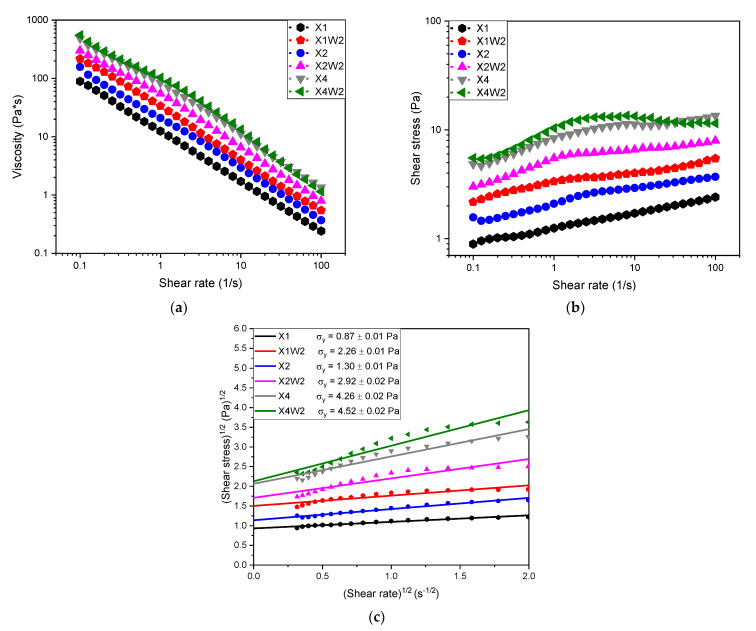
Rheological tests on the hydrogel composites. Trends of (**a**) shear viscosity and (**b**) shear stress as a function of the shear rate. (**c**) Linear interpolation of shear stress values for the determination of the yield stress, according to the Casson model (see Equation (1)).

**Figure 2 molecules-28-01952-f002:**
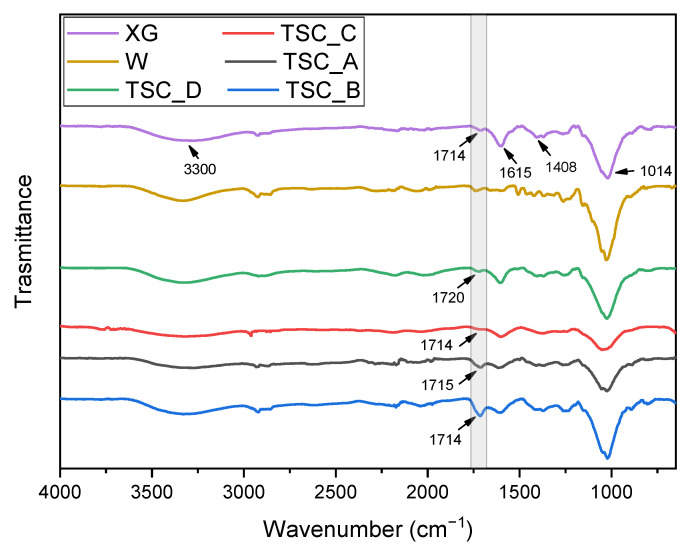
FT-IR analysis on the TSC composites and neat materials.

**Figure 3 molecules-28-01952-f003:**
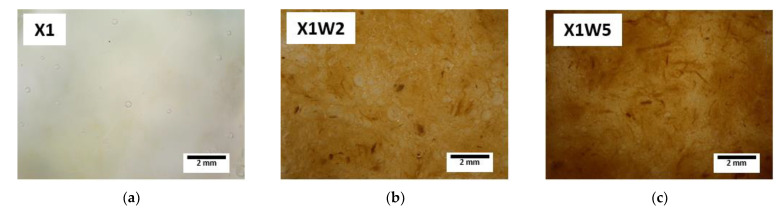
Light microscope images of the prepared hydrogel samples: (**a**) X1, (**b**) X1W2, (**c**) X1W5, (**d**) X2, (**e**) X2W2, (**f**) X2W5, (**g**) X4, (**h**) X4W2, and (**i**) X4W5.

**Figure 4 molecules-28-01952-f004:**
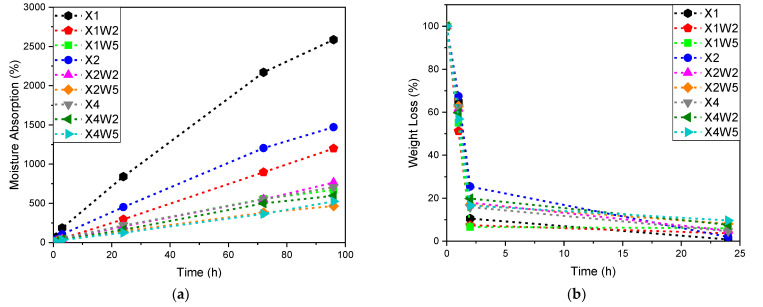
(**a**) Evaluation of the moisture absorption and (**b**) of the water release for the different hydrogel formulations.

**Figure 5 molecules-28-01952-f005:**
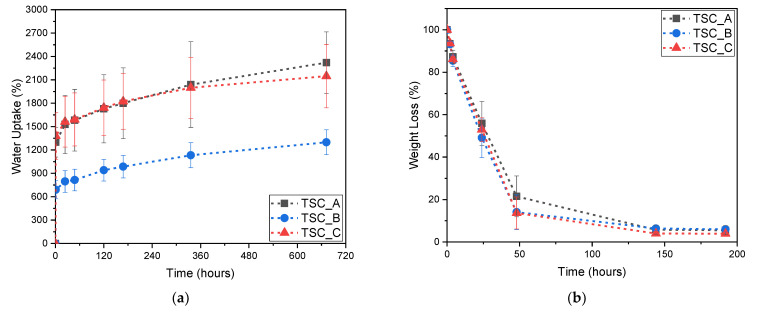
Evaluation of the (**a**) water uptake and of the (**b**) weight loss for the prepared xanthan gum-based TSC.

**Figure 6 molecules-28-01952-f006:**
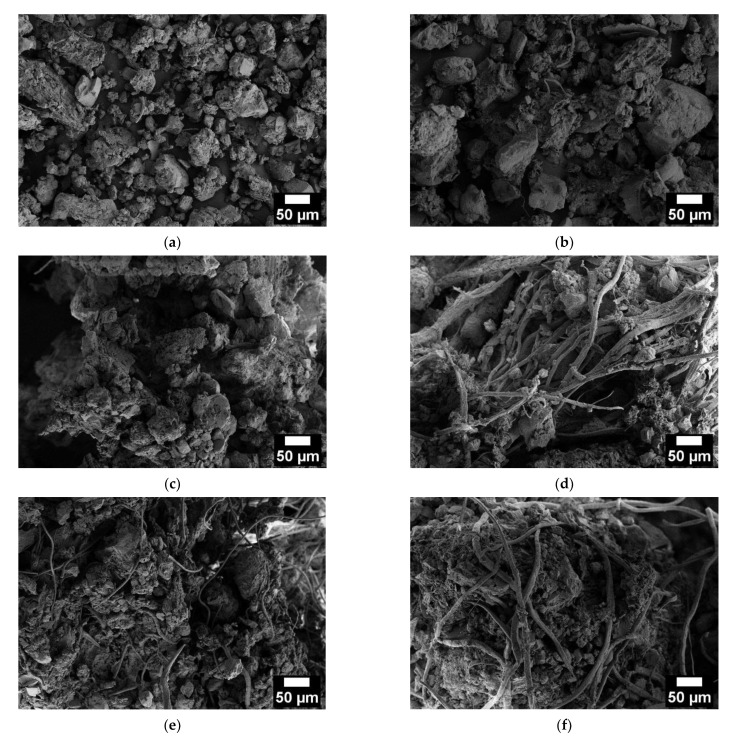
SEM micrographs of the (**a**) neat soil and soil mixed with (**b**) X1, (**c**) X1W2, (**d**) X1W5, (**e**) X4W2, and (**f**) X4W5 hydrogels.

**Figure 7 molecules-28-01952-f007:**
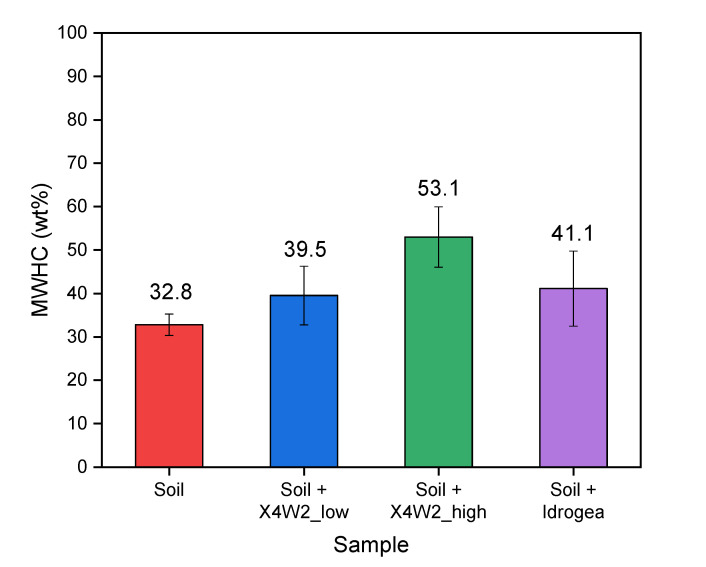
Maximum water holding capacity of soil with X4W2 samples (**low** and **high**) and Idrogea.

**Figure 8 molecules-28-01952-f008:**
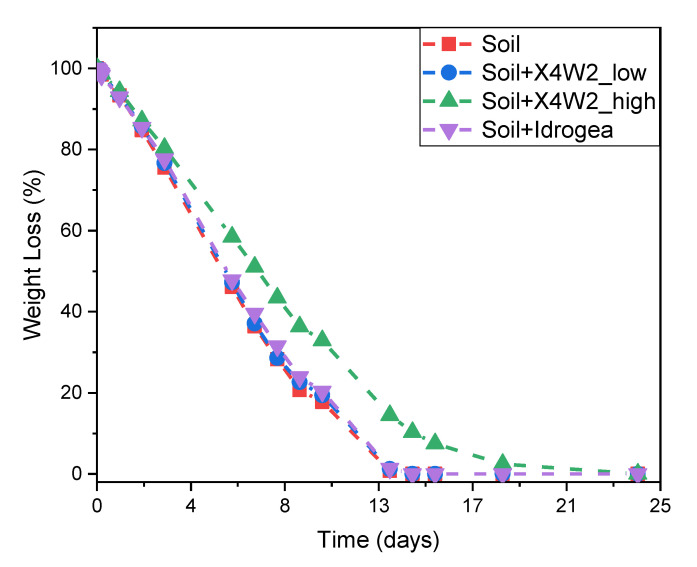
Water retention of the soil with X4W2 hydrogels and Idrogea.

**Figure 9 molecules-28-01952-f009:**
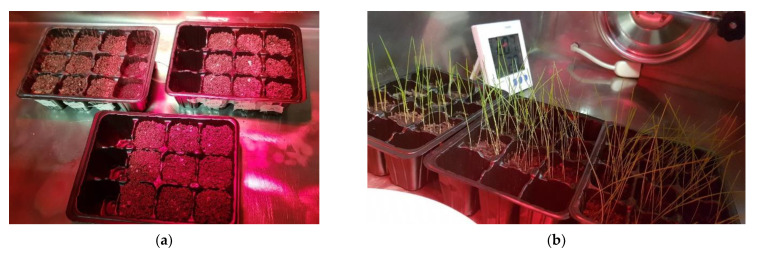
Representative images of (**a**) the grass seeds planted in the different biopolymer-soil culture media and of (**b**) the grass growth after 18 days. (**c**) Germination rate after 18 days and (**d**) average plant height over time.

**Figure 10 molecules-28-01952-f010:**
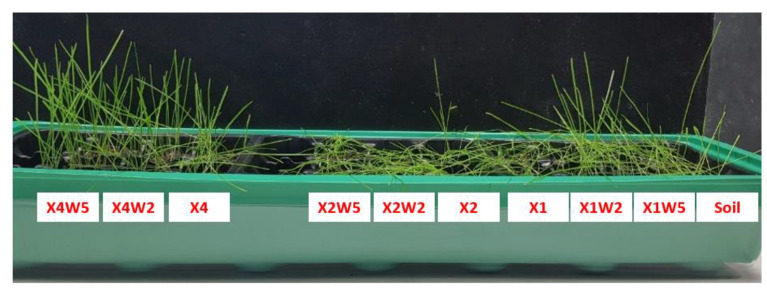
Images of the grass after 7 days of interruption of water supply.

**Figure 11 molecules-28-01952-f011:**
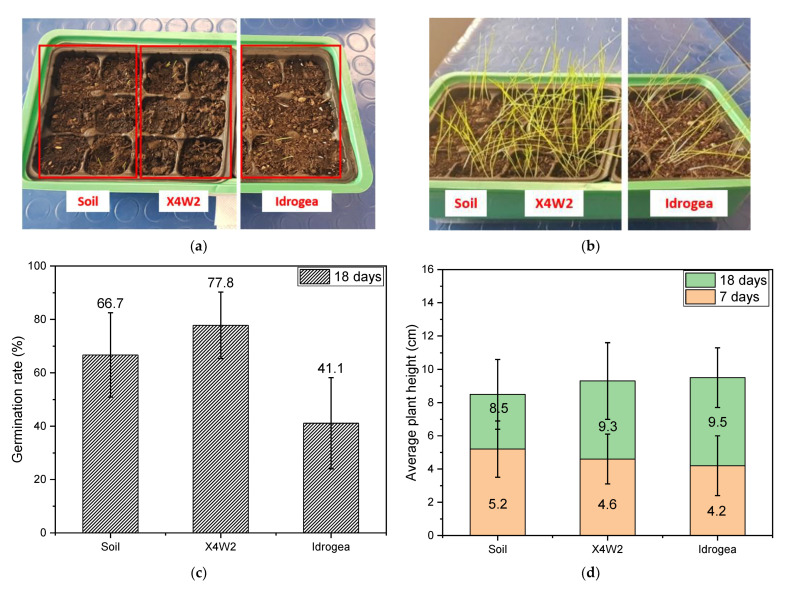
Representative images of (**a**) grass seeds planted in soil, soil treated with X4W2 and with the commercial product, (**b**) grass growth after 14 days. (**c**) Germination rate after 18 days and (**d**) average plant height over time.

**Figure 12 molecules-28-01952-f012:**
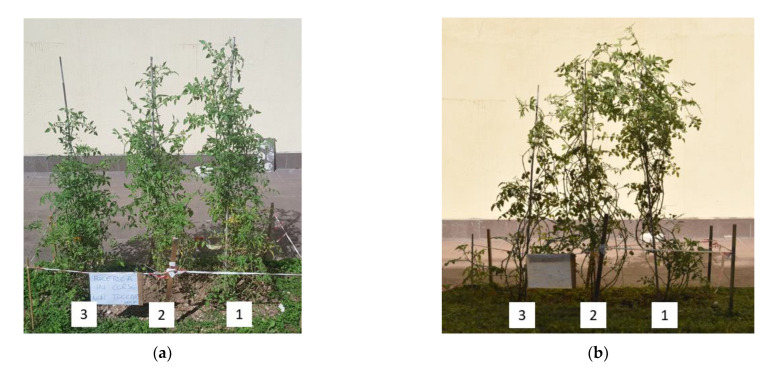
Representative pictures of the plant cultivation with TSC: (**a**) after 60 days and (**b**) after 80 days. (1) not crosslinked TSC, (2) crosslinked TSC, and (3) without TSC.

**Figure 13 molecules-28-01952-f013:**
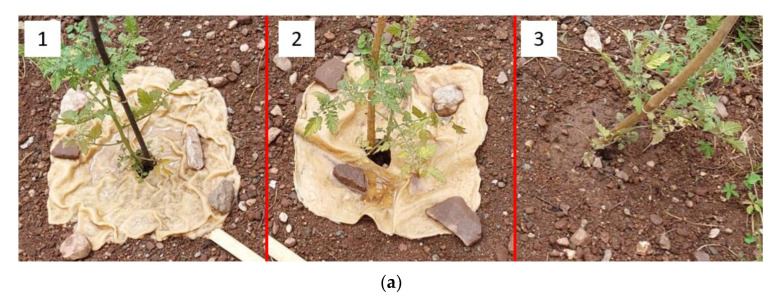
Visual aspect of the TSC (**a**) after 6 days and (**b**) after 80 days. (1) not crosslinked TSC, (2) crosslinked TSC, and (3) without TSC.

**Figure 14 molecules-28-01952-f014:**
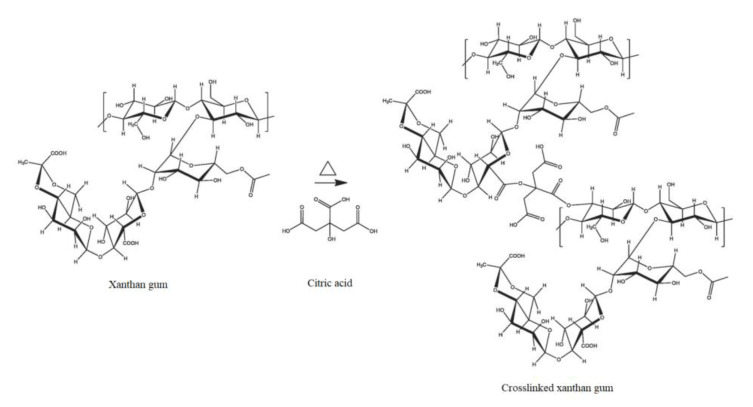
Reaction scheme of the xanthan gum with citric acid (reprinted with permission from [67]).

**Figure 15 molecules-28-01952-f015:**
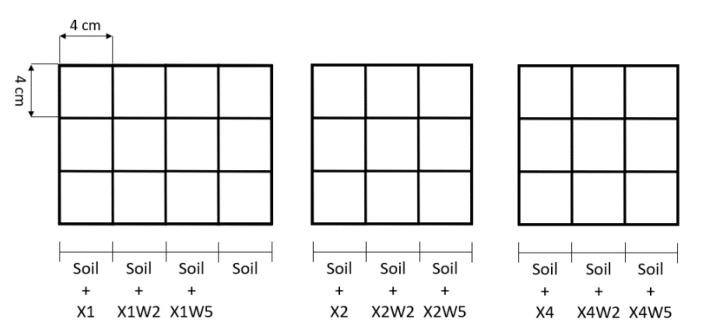
Schematization of the experiment for the evaluation of the grass growth with soil conditioner.

**Figure 16 molecules-28-01952-f016:**
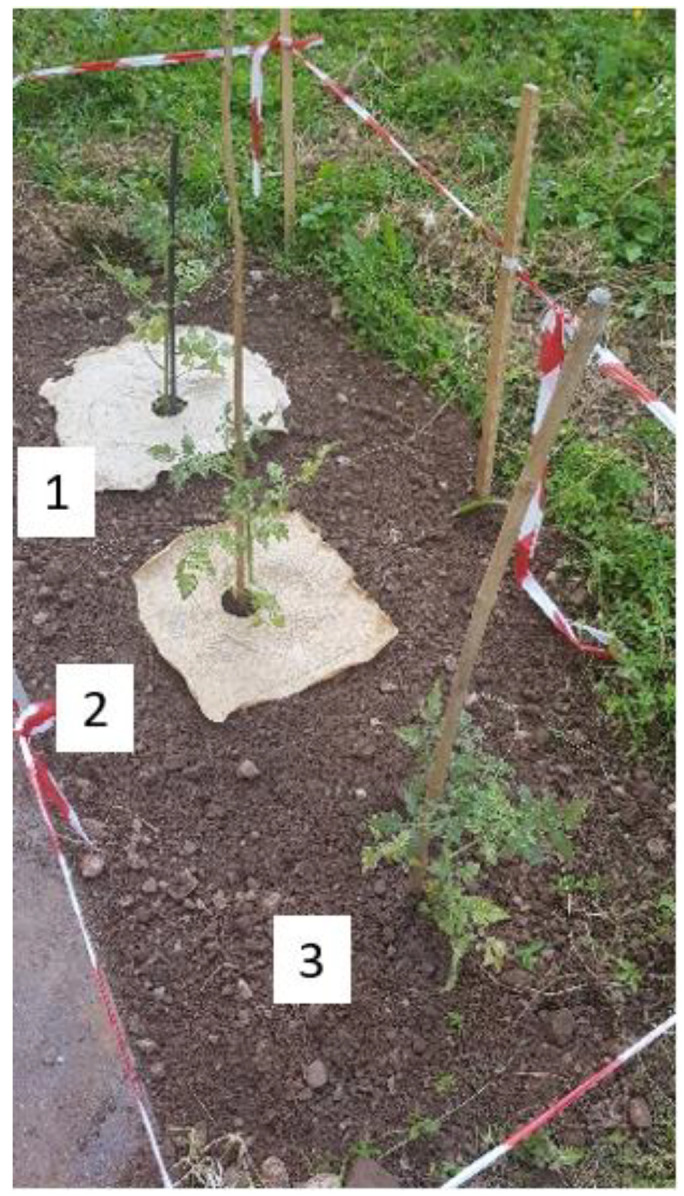
Experimental setup for the tomato plant cultivation with TSC. (1) not crosslinked TSC, (2) crosslinked TSC, and (3) without TSC.

**Table 1 molecules-28-01952-t001:** Results of the chemical analysis of the soil used in this work.

Determination	Value
Sand (2.0–0.05 mm)	595 g/kg
Silt (0.05–0.002 mm)	345 g/kg
Clay (<0.002 mm)	60 g/kg
pH (in water ratio 1:2.5)	7.2
Total Limestone	668 g/kg CaCO_3_
Active Limestone	11 g/kg CaCO_3_
Organic Substance	257 g/kg
Assimilable Phosphorus	45 mg/kg P_2_O_5_
Potassium	112 mg/kg K_2_O
Magnesium	2238 mg/kg MgO

**Table 2 molecules-28-01952-t002:** List of the prepared hydrogel composites, used as SCs.

Sample	Xanthan Gum [wt%]	Cellulose Fibers [wt%]
X1	1.0	0.0
X2	2.0	0.0
X4	4.0	0.0
X1W2	1.0	2.0
X2W2	2.0	2.0
X4W2	4.0	2.0
X1W5	1.0	5.0
X2W5	2.0	5.0
X4W5	4.0	5.0

**Table 3 molecules-28-01952-t003:** List of prepared the xanthan gum-based films, used as TSCs.

Sample	Xanthan Gum [wt%]	Cellulose Fibers [wt%]	Citric Acid [wt%]
TSC_A	4.0	2.0	1.0
TSC_B	4.0	2.0	2.0
TSC_C	4.0	0.0	1.0
TSC_D	4.0	2.0	0.0

## Data Availability

Data are available on request by the corresponding author.

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
