# Peer review of "Development of a Xanthan Gum Based Superabsorbent and Water Retaining Composites for Agricultural and Forestry Applications"

_molecules, 2023, doi:10.3390/molecules28041952_

Round 1

Reviewer 1 Report

The authors have reported xanthan for use in agriculture. the topic is of interest. They have collected decent data, but there are few concerns, which need to be addressed prior to publication. 

1. Cellulose has been used too, a clear cut discussion of xanthan needs to be used alongwith cellulose, and why and how it proved advantageous shud be added

2. The agriculture effect is the best part of the manuscript, but all the results here are qualitative, put up a chart or table, quantifying the height of some factor putting number values to them.

3. Language is not clear in various places , please improve. 

4. section 2 shud be materials and methods that is MDPI format, revise

5. Teresa et al, have reviewed the use of xanthan in agriculture, they have explained many reports, cite those and also what is the novelty factor in your manuscript , please explain  https://doi.org/10.1111/1751-7915.13867 , 

Author Response

Reviewer #1:

  1. Cellulose has been used too, a clear cut discussion of xanthan needs to be used along with cellulose, and why and how it proved advantageous should be added.

Authors thank the referee for the comment. Lines 67-71 of the manuscript have already explained why cellulose fibers have been used and the advantages that cellulose fibers could provide to the xanthan gum hydrogels. However, some references have been added to enforce this concept.

  1. The agriculture effect is the best part of the manuscript, but all the results here are qualitative, put up a chart or table, quantifying the height of some factor putting number values to them.

Authors agree with the referee and Figures 9c, 9d, 10c and 10d with some relative comments have been added to the manuscript.

  1. Language is not clear in various places, please improve.

Authors thank the referee for the suggestion: the manuscript was carefully read and the English improved.

  1. section 2 should be materials and methods that is MDPI format, revise

Authors have checked the instructions provided by the editorial office and for this journal (Molecules) the “Materials and Methods” section should be placed after the “Results and Discussion” section.

  1. Teresa et al, have reviewed the use of xanthan in agriculture, they have explained many reports, cite those and also what is the novelty factor in your manuscript, please explain https://doi.org/10.1111/1751-7915.13867

Authors thank the referee for the suggestion. This review, by Berninger et al., was already known to the authors and indeed was already cited in the manuscript along with some of the works present in this review. Berninger et al. cited many works on the use of xanthan gum for agricultural applications. However, our work can be considered a step forward due to the production of xanthan gum-based composites with the addition of filler (cellulose fibers) that has been shown to have better water retention performances and consequently greater plant growth benefits with respect to the use of xanthan neat hydrogels. Moreover, there are no other studies in the literature regarding the production of bio-based topsoil covers made of crosslinked xanthan gum-cellulose fibers hydrogels.

Reviewer 2 Report

Alessandro Sorze et al. reported “Development of Xanthan gum based superabsorbent and water retaining composites for agricultural and forestry applications”. The paper is publishable after major revising in Polymers journal.

1.       What’s the novelty of the present work because such of bio-hydrogel is already reported? Please mentioned it in revision.

2.       Remove the word “novel” from the abstract part.

3.       The abbreviation “TSCs” replace with full form in the abstract.

4.       Keywords is change with such words which is not mention in the title of the manuscript.

5.       Delete the first sentence from the introduction which is useless.

1.       In the introduction part must explain the different methods that have used to improve water use efficiency and thus to increase the survival rate of growing plants with references. Along with Xanthan gum, other bio-based hydrogels should be included in introduction part. “Acacia Gum Hydrogels Embedding the Insitu Prepared Silver Nanoparticles; Synthesis, Characterization, and Catalytic Application” https://doi.org/10.1007/s10562-020-03380-z, “Highly Versatile Gum Acacia Based Swellable Microgels Encapsulating Cobalt Nanoparticles; An Approach to Rapid and Recoverable Environmental Nano‑catalysis”, https://doi.org/10.1007/s10904-020-01870-6. A Comprehensive Review on Adsorption, Photocatalytic and Chemical Degradation of Dyes and Nitro-Compounds over Different Kinds of Porous and Composite Materials”. Molecules 2023, 28(3), 1081; https://doi.org/10.3390/molecules28031081.

6.       Rephrase the line 88-90. Please be careful during writing.

7.       The result part need to re-write clearly and its English is need improvement.

8.       In Line 109, present and past tens is mixed. Please check it with English expert.

9.       Light microscope images, I cannot see any pores because it 3D polymer and there should be pores in such images. Please explain it with reason.

10.   The increase in the concentration of the biopolymer and cellulose fibers in the solution leads to a decrease in the water absorption capacity. Give the reason here with reference.

11.   SEM images need more explanation.

12.   I have not seen the interaction of cellulose of Xanthan gum. Please include it in the revise manuscript.

13.   FTIR analysis should be included in the revised manuscript because the confirmation of cross-linking will be confirmed through this analysis. FTIR of pure Xanthan gum and cross-linking with cellulose.

14.   The English should be improved with English expert.  

Author Response

Reviewer #2:

  1. What’s the novelty of the present work because such of bio-hydrogel is already reported? Please mentioned it in revision.

Authors thank the referee for the comment. The novelty factors of our work concern the addition of a filler (cellulose fibers) to the xanthan gum hydrogels for the production of composites to be used as bio-based soil conditioners, together with the production of films by crosslinking these xanthan-cellulose composites with a crosslinking agent to be used as bio-based topsoil covers (lines 72-77). In literature there are studies on the use of neat xanthan gum hydrogels for agricultural applications, however in this manuscript the benefits provided by the addition of cellulose fibers to xanthan-based hydrogels for this type of application have been demonstrated.

  1. Remove the word “novel” from the abstract part.

Authors agree with the referee to eliminate the word “novel” from the abstract part.

  1. The abbreviation “TSCs” replace with full form in the abstract.

Authors agree with the referee to replace the abbreviation “TSCs” with the full form in the abstract.

  1. Keywords is change with such words which is not mention in the title of the manuscript.

Authors appreciate the referee's suggestion and have applied some changes. However, we would like to keep some of these keywords because we believe they are representative of the meaning of this manuscript.

  1. Delete the first sentence from the introduction which is useless.

In the introduction part must explain the different methods that have used to improve water use efficiency and thus to increase the survival rate of growing plants with references. Along with Xanthan gum, other bio-based hydrogels should be included in introduction part. “Acacia Gum Hydrogels Embedding the Insitu Prepared Silver Nanoparticles; Synthesis, Characterization, and Catalytic Application” https://doi.org/10.1007/s10562-020-03380-z, “Highly Versatile Gum Acacia Based Swellable Microgels Encapsulating Cobalt Nanoparticles; An Approach to Rapid and Recoverable Environmental Nano‑catalysis, https://doi.org/10.1007/s10904-020-01870-6. A Comprehensive Review on Adsorption, Photocatalytic and Chemical Degradation of Dyes and Nitro-Compounds over Different Kinds of Porous and Composite Materials”. Molecules 2023, 28(3), 1081; https://doi.org/10.3390/molecules28031081.

Authors apologize for the oversight and delete the first sentence of the introduction. The suggested references have been added to the revised manuscript along with others to explain the different methods to improve water use efficiency and increase survival rate of plants.

  1. Rephrase the line 88-90. Please be careful during writing.

Authors thank the referee for the notification and the mistake has been corrected.

  1. The result part need to re-write clearly and its English is need improvement.

Authors thank the referee for the suggestion: the manuscript was carefully read and the English improved.

  1. In Line 109, present and past tens is mixed. Please check it with English expert.

Authors thank the referee for the notification and the mistake has been corrected.

  1. Light microscope images, I cannot see any pores because it 3D polymer and there should be pores in such images. Please explain it with reason.

Authors thank the referee for the comment. The light microscope analysis has been performed on the composite hydrogels in the wet state, therefore the pores are not visible from these images. In future works the analysis will be implemented with images of freeze-dried samples to investigate better their morphology.

  1. The increase in the concentration of the biopolymer and cellulose fibers in the solution leads to a decrease in the water absorption capacity. Give the reason here with reference.

Authors thank the referee for the comment. As described in Section 3.3.4 of the revised manuscript the experiment is indeed a moisture absorption test. For the sake of clarity, in the revised manuscript, the “water uptake measurements” of Section 3.3.4 performed on the hydrogels were changed in “moisture absorption analyses”. Wan et al. (https://doi.org/10.1016/j.compscitech.2009.02.024) found similar results: increasing the amount of fibers in the composite leads to a decrease in moisture absorption capacity. The chemical similarity between xanthan gum and the filler (cellulose fibers) may lead to good interface adhesion between the two components, which, to some extent, can prevent moisture absorbance by slowing down the diffusion of water molecules. In addition, the hydrogels with a higher biopolymer content are more viscous with the possible presence of lumps that may slow down the moisture absorption which is a kinetic diffusion phenomenon.

  1. SEM images need more explanation.

Authors agree with the referee and more explanations have been added in the SEM analysis section.

  1. I have not seen the interaction of cellulose of Xanthan gum. Please include it in the revise manuscript.

Authors apologize but the question is not clear. If the referee is referring to the interaction between the xanthan gum and cellulose fibers, the issue is beyond the aim of this work because the idea of this research is to investigate the feasibility of these composites for agricultural and forestry applications. However, research is currently ongoing on the effect of different types of cellulose fibers on the properties of the xanthan-based composites.

  1. FTIR analysis should be included in the revised manuscript because the confirmation of cross-linking will be confirmed through this analysis. FTIR of pure Xanthan gum and cross-linking with cellulose.

Authors thank the referee for the suggestion. Section 2.2 (in Results and Discussion) and Section 3.3.2 (in Materials and Methods) with FTIR analysis on the samples have been added.

  1. The English should be improved with English expert.

Authors thank the referee for the suggestion: the manuscript was carefully read and the English improved.

Round 2

Reviewer 1 Report

Accept

Reviewer 2 Report

The author addressed all the issues in the revised manuscript and I am going to recommend the manuscript for publication in the present form.